



**GOBLIN: A land-balance model to identify national agriculture and land use pathways to**
**climate neutrality via backcasting**
Colm Duffy[1]*, Remi Prudhomme[2]*, Brian Duffy[6], James Gibbons[3], Cathal O'Donoghue[4],
Mary Ryan[5], David Styles[1]
[1]Bernal Institute, School of Engineering, University of Limerick, Ireland
[2]CIRAD Département Environnements et sociétés, Montpellier, Languedoc-Roussillon, France
[3]School of Natural Sciences, Bangor University, Bangor, Wales, UK
[4]National University of Ireland Galway Policy Lab, University Road, Galway, H91 REW4, Ireland
[5]Rural Economy & Development Programme, Teagasc, Athenry, Co. Galway, Ireland
[6]Independent Researcher
* These authors have contributed equally to this work
*Correspondence to*: Colm Duffy (Colm.Duffy@ul.ie)

17                                        **Abstract**

The Paris Agreement commits 197 countries to achieve climate stabilisation at a global average
surface temperature less than 2°C above pre-industrial times, using nationally determined
contributions (NDCs) to demonstrate progress vis-à-vis this goal. Numerous industrialised
economies have targets to achieve territorial climate neutrality by 2050, primarily in the form
of "net zero" greenhouse gas (GHG) emissions. However, particular uncertainty remains over
the role of countries' agriculture, forestry and land use (AFOLU) sectors for numerous reasons,
*inter alia*: the need to balance mitigation of difficult-to-abate agricultural emissions against
food security; agriculture emissions of methane do not need to be reduced to zero to achieve
climate stabilisation; land use should be a large net sink globally to offset residual emissions.
These issues are represented at a coarse level in integrated assessment models (IAMS) that
indicate the role of AFOLU in global pathways towards climate stabilisation. However, there
is an urgent need to determine appropriate AFOLU management strategies at national level
within NDCs. Here, we present a new model designed to evaluate detailed AFOLU scenarios
at national scale, using the example of Ireland where 34% of national GHG emissions originate
from AFOLU. GOBLIN (General Overview for a Back-casting approach of Livestock
Intensification) is designed to run randomised scenarios of agricultural activities and land use
combinations in 2050 within biophysical constraints (e.g. available land area, livestock
productivities, fertiliser-driven grass yields and forest growth rates). Based on AFOLU
emission factors used for national GHG inventory reporting, GOBLIN then calculates annual
GHG emissions out to 2050 for each scenario. The long-term dynamics of forestry are
represented up to 2120, so that scenarios can also be evaluated against the Paris Agreement
commitment to achieve a balance between emissions and removals over the second half of this
century. We outline the rationale and methodology behind the development of this biophysical
model intended to provide robust evidence on the biophysical linkages across food production,
GHG emissions and carbon sinks at national level. We then demonstrate how GOBLIN can be
applied to evaluate different scenarios in relation to a few possible simple definitions of
"climate neutrality", discussing opportunities and limitations.
*Keywords*: climate policy; climate modelling; LULUCF; GWP; food security; scenario analysis



## 1. Introduction

Article four of the United Nations Framework Convention on Climate Change (UNFCCC) Paris Agreement (UNFCCC, 2015) states that in order for parties to achieve long-term temperature goals, peak greenhouse gas (GHG) emissions must be reached as soon as possible. Parties must strive to "achieve a balance between anthropogenic emissions by sources and removals by sinks of GHGs" (UNFCCC, 2015). The Agriculture Forestry and Other Land Use (AFOLU) sector incorporates both agricultural activities, such as animal husbandry and crop production, and landuse, landuse change & forestry (LULUCF) activities. As such, it contains important GHG sources and sinks, making a net contribution of 24% to global GHG emissions (Smith et al., 2014). However, LULUCF is regarded as a major potential carbon dioxide ($CO_2$) sink that will be central to any future balance between emissions and removals (IPCC, 2019b; Smith et al., 2014). Lóránt and Allen (2019) emphasise the central role that the AFOLU sector will play to reach climate neutrality, through mitigation of current emission sources, reduced emissions intensity of agricultural production linked with increased efficiency, production of bio-based products to substitute more carbon-intensive products, and carbon sequestration.

An increasing number of countries have established ambitious national "climate neutrality" targets for 2050 in legislation (Oireachtas, 2021; Reisinger and Leahy, 2019; UK CCC, 2019). These targets pose a particular challenge for countries with high per-capita GHG emissions and a high percentage land occupation with ruminant livestock production, such as Ireland (Duffy et al., 2020c) and New Zealand (NZ-MftE, 2021) – because of the difficulty of reducing ruminant livestock emissions of methane ($CH_4$) and nitrous oxide ($N_2O$) (Herrero et al., 2016), and the large carbon dioxide ($CO_2$) sinks needed to offset remaining $CH_4$ and $N_2O$ based upon the 100-yr average global warming potentials ($GWP_{100}$) recommended for national inventory reporting (UNFCCC, 2014). Furthermore, meeting climate neutrality targets is likely to require AFOLU sectors to be better than climate neutral – and to provide net GHG offset to compensate for difficult-to-mitigate residual emissions in other sectors, such as aviation (Huppmann et al., 2018).

Hitherto, most national or AFOLU-specific plans for climate neutrality by 2050 have been based on achieving a balance between GHG emissions and removals in terms of $GWP_{100}$ equivalents (Schulte et al., 2013; Searchinger et al., 2021; UK CCC, 2019). However, the warming effect of stable but continuous $CH_4$ emissions is approximately constant, whilst the warming effect of continuous $CO_2$ and $N_2O$ emissions is cumulative (Allen et al., 2018). Consequently, global climate modelling indicates that biogenic $CH_4$ reductions of 24-47%, relative to 2010 are sufficient to achieve climate stabilisation at a global mean surface temperature 1.5 degrees centigrade above pre-industrial times (Rogelj et al., 2018a). A modified version of $GWP_{100}$, termed GWP*, has been proposed to evaluate future climate forcing effect considering the recent _change_ in $CH_4$ emissions, which is more consistent with global climate modelling used to identify climate stabilisation pathways (Huppmann et al., 2018; Rogelj et al., 2018b). However, GWP* diverges from current inventory reporting, and effectively discounts attribution of recent warming caused by existing methane emissions, posing challenges for attribution and questions for international equity if applied to determine climate neutrality at national level (Rogelj and Schleussner, 2019). Furthermore, the Paris Agreement specifically mentions to need to safeguard food security and end hunger (UNFCCC, 2015). Thus, there is considerable debate and uncertainty regarding the broad suite of agricultural and land use activities compatible with climate neutrality at individual country level, strongly depending on GHG aggregation metric (e.g. $GWP_{100}$ or GWP*), and/or various approaches to downscale global emissions and sinks from particular scenarios compatible with





climate stabilisation (Huppmann et al., 2018; Rogelj et al., 2018b), and the particular impacts
of GHG mitigation on food production in different countries (Prudhomme et al., 2021) .There
is an urgent need to explore implications of different definitions for national AFOLU sectors.
Ireland's AFOLU sector provides an excellent case study to explore the implications of
different definitions of, and pathways towards, climate neutrality because it sits at the
international nexus of livestock production and climate mitigation. Agriculture contributes
34% to national GHG emissions (Duffy et al., 2020c) owing to a large ruminant sector
producing beef and milk largely (90%) for international export. Somewhat unusually within
Europe, Ireland's LULUCF sector is also net source of GHG emissions owing to over 300,000
ha of drained organic soils emitting approximately 8 million tonnes of $CO_2$ eq. annually,
compared with a declining forestry sink of approximately 4.5 million tonnes of $CO_2$ annually
(Duffy et al., 2020c). Methane accounts for circa 60% of agricultural GHG emissions, and
LULUCF emissions of $CH_4$ could increase if organic soils are rewetted to reduce $CO_2$
emissions. The future shape of climate neutrality in Ireland's AFOLU sector, and the amount
of beef and milk that can be produced within associated emission constraints, is thus
particularly sensitive to $CH_4$ accounting (Prudhomme et al., 2021). Nonetheless, it is clear that
achieving climate neutrality will require dramatic changes in agricultural and land management
practises, not least because AFOLU emissions have been increasing over the past decade
(Duffy et al., 2020c). The debate about future land use has implications for livelihoods and
cultural norms (Aznar-Sánchez et al., 2019), and is therefore highly sensitive. In such a context,
pathways to climate neutrality cannot be objectively identified through extrapolation of recent
trajectories nor stakeholder "visions", invoking the need for a backcasting approach to first
establish what a climate neutral AFOLU sector *could* look like.
This paper presents a new biophysical model capable of identifying broad pathways towards
climate neutrality in Ireland's AFOLU sector, "GOBLIN" (General Overview for a Back-
casting approach of Livestock Intensification). GOBLIN integrates, with sensitivity analyses,
key parameters that influence agricultural production, GHG fluxes, ammonia ($NH_3$) emissions
and nutrient losses to water, using methodology aligned with Ireland's UNFCCC reporting.
The model is designed to be run repeatedly with randomly varied, biophysically compatible
combinations of parameter inputs in order to identify specific combinations of agricultural
production and land use that achieve climate neutrality from 2050 through to 2120. In the
following sections, we will describe the scope, model architecture, implementation and
functionality of GOBLIN, ending with discussion on its suitability for intended application and
conclusions.

## 2. Model scope & description

The scope of GOBLIN is currently confined to national AFOLU boundaries (Fig. 1),
accounting for the main AFOLU sources and sinks reported in national inventory reporting
(Duffy et al., 2020), *inter alia,* $CO_2$ fluxes to and from (organic) soils and forestry, $CH_4$
emissions from enteric fermentation, manure management and wetlands, and direct and indirect
losses of nitrogen (N) from animal housing, manure management and fertiliser application, in
the form of $N_2O$, ammonia ($NH_3$) and dissolved forms (e.g. nitrate, $NO_3$) (Duffy et al., 2020).
Fig. 1 highlights the main sources and sinks accounted for in GOBLIN, alongside related
sources and sinks that will be accounted for in subsequent life cycle assessment (LCA) through
coupling and/or integration with related models (Forster et al., 2021; Soteriades et al., 2019;
Styles et al., 2016, 2018).






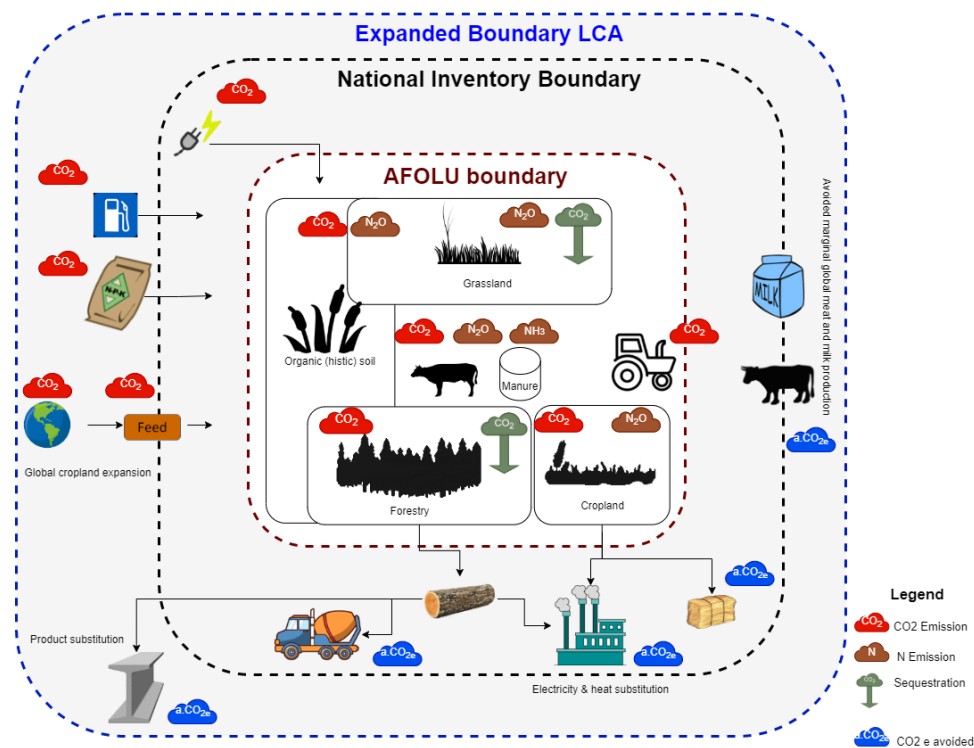


**Figure 1.** **Key emissions sources and sinks critical to the determination of "climate neutrality" in Ireland's AFOLU sector accounted for in GOBLIN (white), alongside linked upstream- and downstream- sources and sinks to be included in subsequent life cycle assessment (LCA) modelling to determine wider climate mitigation efficacy.**

In the form of a global sensitivity analysis (Saltelli et al., 2009), GOBLIN varies key uncertain
parameters within the AFOLU sector to calculate emissions and sequestration up to the year
2120, associated with linear rates of land use change up to the initial target year for neutrality,
2050 (additional complexity around forestry described later). The back-casting approach used
in GOBLIN makes explicit the linkages across biophysical constraints relating model outputs
(emission reduction targets) with model inputs (parameters defining production systems and
land management). These explicit linkages enable GOBLIN users to better understand
complementarities and trade-offs across AFOLU activities with respect to the climate neutrality
objective, based on transparent and objective scenario construction. A primary aim of the
model is to ensure consistency of scenarios in terms of land use (e.g. within available areas for
grazing and carbon sequestration), associated agricultural production potential within land
constraints and related to key production efficiency parameters, and associated GHG fluxes.
The model  allows scenarios to be built based on standardized sampling methods for key input
parameters, avoiding sampling bias introduced by screening methods (Saltelli et al., 2000). The
model is designed to run a large number (e.g. 100s) of times to generate a suite of results
representing different land use scenarios by 2050, and time series of emissions and
sequestration up to 2120. Scenarios can then filtered to identify which ones comply with





climate neutrality based on different definitions and metrics, e.g.: (i) net zero GHG balance
based on $GWP_{100}$ (IPCC, 2013); (ii) no *additional* warming based GWP* (Allen et al., 2018;
Lynch et al., 2020); (iii) compliance with a specific $CH_4$ targets downscaled from Integrated
Assessment Models (IAMs) combined with a $GWP_{100}$ balance across $CO_2$ & $N_2O$ fluxes.
Climate neutrality can be determined at one point in time (2050), and/or as a time-integrated
outcome over the second half of the century as per the Paris Agreement (UNFCCC, 2015).
Filtered scenarios enable identification of input combinations compatible with climate
neutrality as an objective evidence base for stakeholders to elaborate more detailed pathways
towards climate neutrality considering wider socio-economic factors (Clarke et al., 2014).
A key feature of GOBLIN is its relation of complex interactions across livestock production,
grassland management and emissions offsetting within the AFOLU sector to a few simple input
parameters used to define a plethora of possible scenarios. Reflecting the dominance of bovine
production within Ireland's AFOLU sector, primary input data to initialise the model are
national herd sizes (derived from milking cow and suckler-cow numbers) and average animal-
level productivity (e.g. milk yield per cow) to determine feed energy intake, fertiliser
application rates and grass utilisation rates to determine stocking densities and production
outputs, followed by proportions of any spared grassland (relative to the baseline year) going
to alternative land uses. In v1.0, alternative land uses are limited to fallow or commercial or
conservation forestry and rewetting of drained organic soils (bioenergy cropping and anaerobic
digestion can be readily integrated for coupling with downstream energy models). Subsequent
iterations and model coupling will account for upstream effects of e.g. fertiliser and feed
production and extend downstream value chains to consider e.g. energy and material
substitutions, taking a full LCA approach (Fig. 1). Activity data and emission coefficients are
largely based on those used in Ireland's National Inventory Report (NIR) (Duffy et al., 2020),
which are in turn based on IPCC (2006) and IPCC (2019a) good practice guidelines for national
GHG reporting at Tier 1 level for soil emissions, Tier 2 level for animal emissions and Tier 3
level for forestry carbon dynamics.
**2.1 Modelling architectural overview**
GOBLIN incorporates seven modules, displayed in a dataflow diagram (Pressman, 2010) in
Fig. 2, a number of which are derived from previous models on national grassland
intensification (Mc Eniry et al., 2013), farm LCA (Jones et al., 2014; Styles et al., 2018) and
forest GHG fluxes (Duffy et al., 2020a). The flow of data is represented by arrows between
interlinked modules (brown rectangles), processes (purple circles) and data stores (green, open
ended rectangles) (Fig. 2). The scenario, herd, grassland, livestock, land use, forestry, and
integration modules included in GOBLIN reflect initiation and synthesis functions, along with
data on the main activities and emissions arising within the AFOLU sector. The modules are
run in sequential order, with subsequent modules relying on the output generated by previous
modules.
Initially, the scenario generation module (1) varies the key input parameters utilised in the sub
modules. The cattle and sheep livestock herd module (2) computes the national cattle herd and
ewe flock from milking and suckler cow numbers and upland and lowland were numbers (input
parameters) based on coefficients derived from the average national composition (Donnellan
et al., 2018) – see Table 3. The grassland module (3) computes the energy (feed) requirements
of each animal cohort within the national herd, fertiliser application and subsequently the area
of grassland needed (depending on concentrate feed inputs, fertiliser application rates and grass
utilisation rate) and the grassland area free for other purposes ("spared grassland"). Emissions
related to livestock production are computed in the livestock module (4) and rely on inputs



from the cattle herd (2) and grassland (3) modules, based on a Tier 2 IPCC approach (Duffy et
al., 2020c; IPCC, 2019a). Once the grass and concentrate feed demand has been calculated
(detailed in subsequent sections), using the herd and grassland modules, the land use module
(5) computes the remaining emissions from land uses related to cropland, wetlands, settlements
and other land. The remaining LULUCF categories related to forest are captured in the forest
module (6) and are utilised by the land use module (5). The scenario generation module
provides the proportion of spared grassland to be converted to each alternative land use
(forestry, rewetting, etc.). GOBLIN does not currently include a harvested wood products
module, however, this will be included in subsequent versions, and will utilise harvestable
biomass outputs from the forest module related tree cohort, management practises and age
structure. The sequential resolution of these modules allows for an accurate representation of
biophysically resolved land use combinations in terms of land areas, production (meat, milk,
crops and forestry) and emissions.





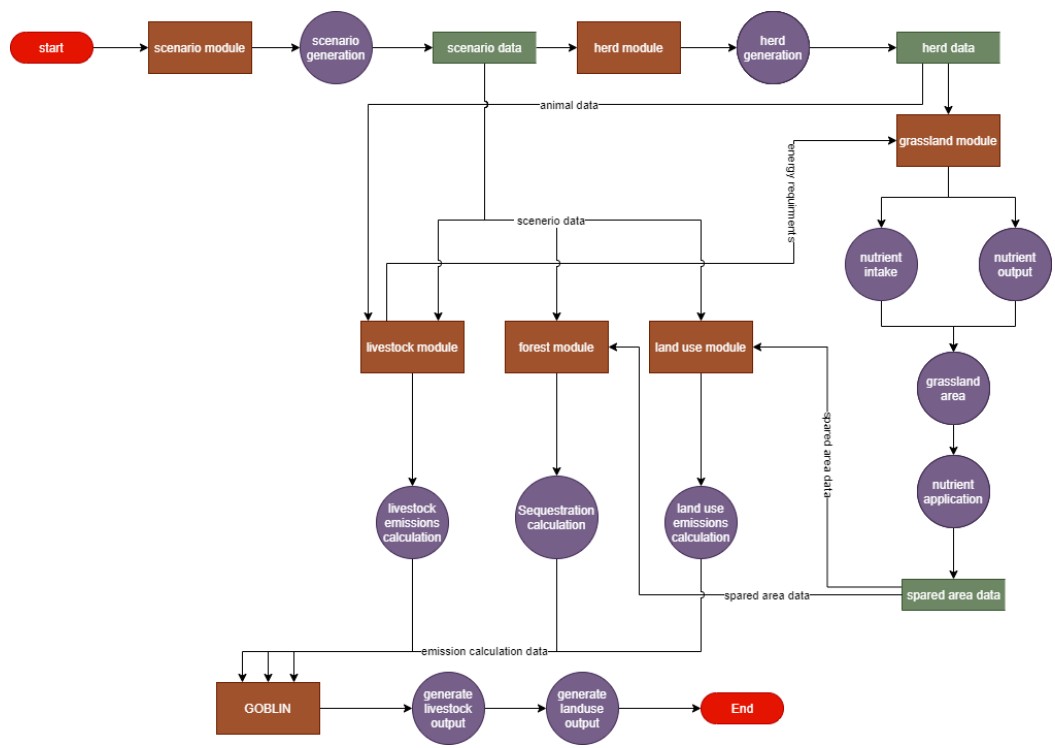


**Figure 2.**    **GOBLIN Data Flow Diagram. Arrows represent data flow. Modules are represented by brown rectangles, processes by purple circles, and open-ended green rectangles represent data stores.**

## 2.2 Modelling Application

Grass feed requirements are calculated based on the Tier 2 IPCC (IPCC, 2006) net energy requirements for livestock ($NE_{feed}$) related to animal cohort (c) and productivity (p), minus net energy received from supplementary (concentrate) feeds ($NE_{supp.}$) and grass net energy density ($D_{NE\text{-}grass}$) (Eq. 1). Subsequent calculation of N excretion ($N_{ex}$) from animals and share of time indoors (IPCC, 2019a) enables average organic nutrient loading to grassland to be calculated. Organic nutrient loading is then combined with average synthetic fertiliser application rate (exogenous variable) to determine total N inputs ($N_{input}$) and average grass yield ($Y_{grass}$) based on the grass yield function reported by Finneran et al. (2012). According to the grass utilisation coefficient ($U_{grass}$), calibrated for baseline (2015) animal grass feed requirements and grassland area ($A\text{-}BL_{grass}$), the calculated required area of grassland is then subtracted from the grassland area reported in the baseline year (2015) to calculate spared grass area ($A\text{-}S_{grass}$).







$$A - S_{grass} = A - BL_{grass} - SUM_{c,p}\left(\frac{\frac{NE_{feed} - NE_{supp}}{D_{NE-grass}}}{Y_{grass} \cdot U_{grass}}\right) \tag{1}$$


Spared grassland area is then apportioned to various alternative land uses based on exogenous
inputs via the scenario module. The GOBLIN integration module then combines outputs from
the grassland, livestock, forest and land use modules to calculate relevant GHG fluxes. Table
1 gives a brief description of the modules and their purpose. The following sections will
elaborate on scenario generation, cattle herd building, grassland management, land balance,
emissions and forestry sequestration calculations.
**Table 1.        Summary of module functions within GOBLIN**

| Module | Function | Details |
|---|---|---|
| Scenario Module | The production of randomised scenario parameters. | Samples input variables from predefined maximum ranges (technical potential) with a Latin Hyper Cube algorithm to build each of the scenarios. |
| Herd Module | The generation of dairy, cattle, upland and lowland sheep national herd/flock numbers. | Utilises herd/flock coefficient data derived from (Donnellan et al., 2018) to create the national herd based on milking- and suckler- cow numbers and ewe numbers (from Scenario module). |
| Grassland Module | Calculation of grassland area required for livestock production and calculation of nutrient application to grassland area. | Utilises IPCC (IPCC, 2006) guideline tier 2 functionality to calculate grass land area required based on: (i) nutritional requirements of the national herd (see Eq. 1); (ii) organic N returns to soil; (iii) average fertiliser application rates, linked with grass productivity fertiliser response curve.<br><br>Deduces spared grassland available for other purposes (Eq. 1). |
| Livestock Module | Calculation of agricultural emissions and nutritional requirements related to livestock production. | Algorithms for emissions of $CH_4$, $N_2O$, $NH_3$ and $CO_2$ to air based on IPCC (IPCC, 2006) and IPCC (IPCC, 2019a) methodologies.<br><br>Includes tier 2 functionality for the estimation of nutritional requirements of livestock. |
| Land Use Module | Calculation of emissions related to land use and land use change | Algorithms for emissions of methane $CH_4$, $N_2O$, $NH_3$ and $CO_2$ to air based on IPCC (IPCC, 2006) and IPCC (IPCC, 2019a) methodologies.<br><br>Land use calculations relate to forested lands, wetlands and grasslands. |





| Forestry Module | Calculation of emissions and sequestration related for afforestation. | Calculation of forest sequestration based on IPCC (IPCC, 2006), IPCC (IPCC, 2019a) and Duffy et al (Duffy et al., 2020a). Past sequestration is estimated as well as projected future sequestration. Other emissions associated with management of soils under forestry are also calculated here. |
| GOBLIN Module | Coordination and integration of the program modules and production of final results. | Management module utilising tools and functions from previous modules to produce the final results. |


### 2.2.1 Scenario Generation

There are 65 input parameters included in the global sensitivity analyses that influence the outputs of GOBLIN. Table 2 outlines the definitions, baseline values and scenario ranges of the key input parameters. The objective of the GOBLIN model is to identify which combinations of input variables are compatible with climate neutrality in the target year. With this number of input parameters (65) and the complexity of the relationships between them, it is impossible to study all combinations of parameters. To reduce the number of simulations while keeping a broad and unbiased exploration of the possible value ranges for these parameters, a Latin Hypercube sampling algorithm will be employed (McKay et al., 2000). This established sampling method allows the values taken by the input parameters in the scenarios to be distributed across plausible (technically possible) ranges.

**Table 2.    Definitions and selected value range examples for key GOBLIN input parameters**

| Parameter category | Definition | Baseline (2015) values | Scenario value range |
| --- | --- | --- | --- |
| Livestock population | Milking cow/suckler-cow/sheep numbers | • Milking cow: 1,268,000<br>• Dry cow: 1,065,000<br>• Lowland ewe: 1,960,000<br>• Upland ewe: 490,000 | • Milking cow: 0 – 1,430,000<br>• Dry cow: 0 – 1,550,000<br>• Lowland ewe: 0 – 1,960,000<br>• Upland ewe: 0 – 440,000 |
| Productivity | Milk and beef output per head | • Milk output: 13.8 kg per cow per day<br>• Beef finish weights for heifer 1 & 2 years: (275, 430 kg per head) | • Milk output: 13.8 – 15.9 kg per cow per day<br>• Beef finish weights for heifer 1 & 2 years: (275, 430 kg per head) - (322, 503 kg per head) |
| Grassland area | | 4.07 M ha | Deduced |
| Cropland area | | 361.6 k ha | Static |
| Drained organic grassland soils | | 287 k ha | Deduced from spared grassland area |
| Wetland area | | 1226 k ha | Deduced |
| Drained wetland area | | 63 k ha | Deduced |





| Grassland utilisation | The proportion of grass production consumed by livestock via grazing and feeding on conserved grasses (silage and hay). | 57% | 50% – 80% |
|---|---|---|---|
| Afforested area | The proportion of spared grassland area on mineral soils that will be utilised for forest. | NA | 0 – 100% of spared mineral soil area |
| Proportion broadleaf | Proportion of forest area that is under broadleaf (vs conifer). | 20% (existing forest) | 30% – 100% (new forest) |
| Proportion conifer harvested | Proportion of conifer area that is harvested. | 90% (existing forest) | 0 – 100% (new forest) |
| Proportion of conifer thinned | The proportion of harvested conifer area that is thinned. | 50% (existing forest) | 0-100% (new forest) |

These input parameters are randomly varied and then utilised by downstream modules to
generate results.

**2.2.2 Cattle herd model**

Calculation of national livestock numbers relies on coefficients relating animal cohorts to the
numbers of milking- and suckler-cows (Donnellan et al., 2018). In terms of cattle production,
dairy (milking) and beef-suckler cow numbers are exogenous parameters bounded between 0
and 1.43 and 0 and 1.55 million head respectively in each scenario. A calving rate of between
0.81 and 1 for dairy cows, and between 0.8 and 0.9 for suckler cows, is used to derive the
number of 1st year and second year male and female calves (48 % of male calves under 1 year,
44% of male calves between 1 and 2 years and 46% of male calves over 2 years). The dairy
and suckler heifers are then derived with a replacement rate of respectively 0.23 and 0.15.
Finally, the number of bulls is computed as a share of suckler cows. The dairy and beef herd
are thus recomputed for different dairy and suckler cow numbers. Table 3 shows the
coefficients utilised in the computation of national cattle and sheep herds for 2015, based on
the number of milking, suckler cows, and upland and lowland ewes.

**Table 3.     Coefficients utilised to compute animal numbers across cohorts based on**
**milking- and suckler-cow numbers**

| Livestock System | Goblin Animal Cohorts | Value |
|---|---|---|
| Dairy & Beef | Heifer aged more than two years | 0.22 |
| Dairy & Beef | Heifer aged less than two years | 0.59 |
| Dairy& Beef | Male calves | 0.44 |
| Dairy& Beef | Female calves | 0.44 |
| Dairy & Beef | Steers | 0.27 |
| Dairy & Beef | Bulls | 0.01 |
| Sheep | Lowland lamb aged more than one year | 0.06 |





| Sheep | Lowland lamb aged less than one year | 0.45 |
| Sheep | Male lowland lamb aged less than one year | 0.45 |
| Sheep | Lowland ram | 0.03 |
| Sheep | Upland lamb aged more than one year | 0.06 |
| Sheep | Upland lamb ages less than one year | 0.45 |
| Sheep | Male upland lamb aged less than one year | 0.45 |
| Sheep | Upland lamb | 0.031 |

*Animal cohort populations are calculated as a proportion of adult stock utilising the relevant cohort coefficient,
derived from Donnellan et al (Donnellan et al., 2018).
Estimation of current milk yield is derived from CSO (2018) and future milk yield are based
on the Teagasc (Teagasc, 2020b) dairy sector roadmap. The milk yield ranges from 5049 to
5800 kg of milk per cow per year. Live weights are based on research conducted by O'Mara et
al (O'Mara, 2007). Live weight gain of female and male calves are kept constant at 0.7 and 0.8
kg/head/day, respectively, and average baseline live weights for dairy cattle are assumed
constant at 538, 511, 300, 290, 320 and 353 kg/head for milking cows, dry cows, heifers, female
calves, male calves and bullocks, respectively, based on farm LCA model default values
(Soteriades et al., 2018). The same is assumed in relation to beef cattle with the exception year
1 and 2 heifers whose live weights range from 275 to 322 and 430 to 503kg/head, respectively.
Increased beef liveweights are based on the Teagasc sectoral roadmap (Teagasc, 2020a). Live
weights, live weight gains and milk yields, are used to calculate net energy requirements for
specified animal cohorts (IPCC, 2006).
**2.2.3 Grassland management module**
The purpose of the grassland module is to estimate the required area of land necessary to
maintain the scenario-specific herd/flock at a given yield and utilisation rate. Grassland
utilisation rate is calibrated at 57% based on calculated grass uptake and total grassland area
utilised in baseline year (2015). The calibrated rate is between the average rate of 60% reported
by McEniry et al. (2013), and a rate of 53% deduced from average grass dry matter (DM)
utilisation report by Creighton et al. (2011) divided by average DM production reported by
Donovan et al (2021). The estimation of grassland area is contingent on establishing the energy
requirements of herd/flock and grassland fertilisation rates, as described above. Fig. 3 shows
the data flow within the grassland module.






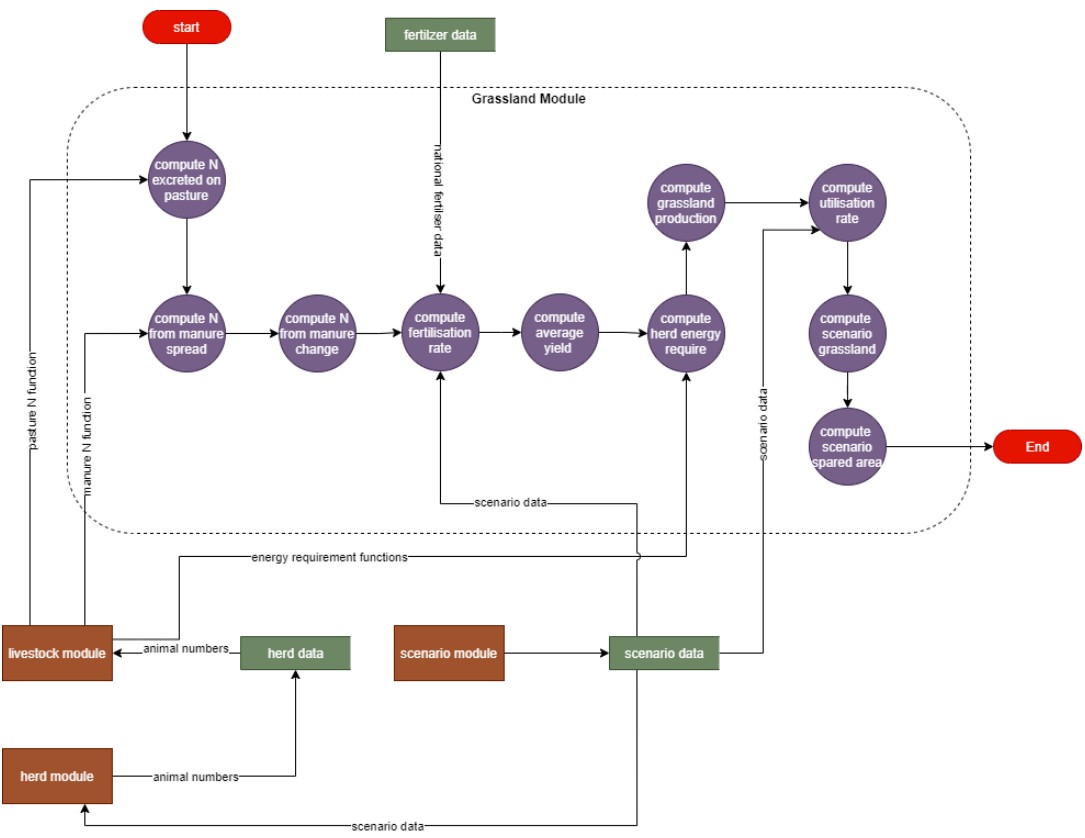

**Figure 3.** **Data flow and processing through the grassland module. Arrows represent**
**data flow. Modules are represented by brown rectangles, processes by purple circles,**
**and open-ended green rectangles represent data stores.**
Grassland production is computed per major soil group (Gardiner and Radford, 1980; McEniry
et al., 2013), from group 1 (highest productivity potential) to group 3 (lowest productivity
potential). Each grass type has a different yield class (YC) based on its soil group. GOBLIN's
grassland module deduces the area required to satisfy the livestock grass demand for each
category of grass (pasture, silage, hay) for each YC (1,2,3) and year. The basic equation is as
follows:

$$D_{land,grass,YC,t} = \frac{S_{grass,YC,t}}{Y_{grass,YC,t}} \tag{2}$$

Where $D_{land}$ refers to area demand, $grass$ refers to grass type, $YC$ refers to grass YC based
on soil group, and $t$ refers to year. The parameter $S_{grass}$ refers to the grass supply, while $Y_{grass}$
refers to the grass yield.



GOBLIN allocates the silage, hay and grazed grass requirement at the year t ($S_{grass,t}$) between
soil group based on the share the soil group in the grass production at the reference year (2015)
($\frac{S_{grass,YC,2015}}{S_{grass,2015}}$) as following:

$$S_{grass,YC,t} = S_{grass,t} \times \frac{S_{grass,YC,2015}}{S_{grass,2015}} \qquad (3)$$

The grassland management module utilises a similar approach to the determination of grassland
DM yield reported by McEniry et al. (2013), based on Finneran et al (2011):

$$Y_{grass,YC,t} = f(N_{rate}) \times \propto_{yield\ gap,YC} \times \propto_{Utilisation,t} \qquad (4)$$

Where $f(N_{rate})$ refers to the maximum yield response to fertiliser nitrogen rate from Finneran
et al. (Finneran et al., 2012) in experimental fields, given as:

$$f(N_{rate}) = -0.000044 \cdot N_{rate}^2 + 0.038 \cdot N_{rate} + 6.257 \times \frac{N_{rate}^{manure}}{N_{rate,ref}^{manure}} \qquad (5)$$

where $N_{rate}^{manure}$ is the manure excretion on pasture and $N_{rate,ref}^{manure}$ is the manure excretion on
pasture in the reference year. This term considers the influence of the livestock stocking rate
on pasture fertilization. For grassland other than pasture (Hay and grass silage), $\frac{N_{rate}^{manure}}{N_{rate,ref}^{manure}} = 1$.
$N_{rate}$ represents the nitrogen application (manure and synthetic application).
The remaining elements of equation 4 are $\propto_{yield\ gap,YC}$ and $\propto_{Utilisation,t}$, where $\propto_{yield\ gap,YC}$
refers to the yield gap of each YC category (0.85, 0.8 and 0.7 for respectively YC 1,2,3), and
$\propto_{Utilisation,t}$ refers to the utilisation rate (calibrated as described above).
Once land use demand has been satisfied, the area available for land use change
($D_{land,available}$) is computed as follows:

$$D_{land,available} = \sum_{grass,YC} D_{land,grass,YC,2015} - D_{land,grass,YC,t} \qquad (6)$$

Once the spared area ($D_{land,available}$) has been determined, it can then be allocated to
alternative land uses.

## 3. GHG fluxes

The GOBLIN integration module coordinates the livestock and other agricultural emissions
with LULUCF fluxes. The following subsections will elaborate on each of these in turn,
beginning with the estimation of livestock and other agricultural emissions

### 3.1 Livestock emissions

This module utilises an adapted farm LCA model developed in previous studies of UK
livestock systems (Soteriades et al., 2018, 2019b; Styles et al., 2015) to estimate environmental



footprints. Algorithms for emissions of $CH_4$, $N_2O$, ammonia ($NH_3$), and $CO_2$ to air were applied to relevant activity data inputs. Enteric $CH_4$ and manure management $CH_4$ and $N_2O$ emissions were calculated using IPCC Tier 2 equations (IPCC, 2006, 2019a) and Tier 2 calculation of energy intake and $N_{ex}$ according to dietary crude protein (CP) intake. Enteric fermentation is based a methane conversion factor ($Y_m$) value of 6.5%, and 4.5% for lambs, applied to gross energy intake calculated by cohort as previously described, and an average feed digestibility of 730 g/kg for Irish cattle (Duffy et al., 2020c). Soil $N_2O$ emissions are derived from $N_{ex}$ during grazing, and the application of synthetic fertiliser (as urea or calcium ammonium nitrate) and manure spreading. Indirect emissions of $N_2O$ were calculated based on $NH_3$ emission and N-leaching factors from the most recent national emission inventory (Duffy et al., 2020c).

Emissions of $CH_4$, $NH_3$ and direct/indirect $N_2O$ from housing and manure management were calculated from total $N_{ex}$ indoors based on the proportion of time animals are housed, housing type, and manure management system specific emission factors (IPCC, 2019). The fraction of time spent indoors for milking cows, suckler cows, heifers, female and male calves, bullocks and bulls are respectively, 0.43, 0.39, 0.36, 0.48, 0.07 and 0.43 (O'Mara, 2007). Manure storage $NH_3$-N EFs of 0.05 and 0.515 of total ammoniacal N (TAN) for tanks (crusted) and lagoons were taken from (Misselbrook et al., 2010), assuming 60% of N excretion is TAN (Webb and Misselbrook, 2004) – applied to 92% and 8% of managed cattle manures, respectively (O'Mara, 2007).

## 3.2 Soil emissions

Emissions from agricultural soils originate from mineral fertilization, manure application and urine and dung deposited by grazing animals. The average annual mineral N fertilization rate across all grassland is 70 kg ha$^{-1}$ in the baseline (McEniry et al., 2013). Direct $N_2O$ emissions for manure spreading are calculated based on IPCC (IPCC, 2006) utilising an emissions factor of 0.01 kg $N_2O$-N/kg N. The NIR (2020c) utilises country specific disaggregated emissions factors in relation to direct emissions from faeces and urine, which are 56% lower than that of the IPCC (2006). As such, an emissions factor of 0.0088 is utilised for urine and dung deposits. An assumption of 10% leaching of fertiliser, residue and grazing N inputs to water is also utilised (Duffy et al., 2020). In addition, an $NH_3$-N emissions factor of 0.06 was applied to grazing TAN deposition (Misselbrook et al., 2010). Indirect $N_2O$-N emissions were calculated as per (IPCC, 2019a): 0.01 of volatilized N, following deposition, and 0.01 of leached N. Other sources (residues, cultivation of organic soils, mineralization associated with loss of soil organic matter) are kept constant. NIR (2020c) country specific emissions factors relating to synthetic fertiliser direct emissions were applied. These emissions factors correspond to: 0.014, 0.0025 and 0.004 kg N2O-N/kg N applied, respectively for CAN, urea and urea + n-butyl thiophosphoric triamide. The fraction of synthetic fertiliser N that volatilises as NH3 and NOx (kg N volatilised (kg of Napplied)$^{-1}$) is also disaggregated by type (0.45, 0.097 and 0.02 corresponding to urea, urea + n-butyl thiophosphoric triamide and CAN, respectively). These values are based on updated IPCC Misselbrook and Gilhespy (2019).

Emissions from organic and mineral grassland area are computed utilising a IPCC (2006) Tier 1 methodology. Areas of mineral soil under improved, unimproved and rough grazing grasslands and areas of organic soil under different management are deduced from the NIR of 2017 (Duffy et al., 2018). The $CO_2$ emissions from land-use change on mineral soils between grassland and other land uses are based on IPCC (2006) methodology. Emissions of $CH_4$, $N_2O$ and $CO_2$ from organic soils are computed for drained and rewetted soils based on the Tier 1 of IPCC methodology described in the 2013 wetlands supplement (Hiraishi et al., 2014).



### 3.3 Land use module

The land use module coordinates a range of emission calculations and allocation of spared land between different land uses based on input parameters defined in the scenario module, as outlined in the subsections below.

### 3.3.1 Land-use allocation

Spared land is computed in the grassland module. The proportion of spared area that is organic or mineral soil is defined by the scenario input parameters. The proportion of area that is organic is limited by the total organic grassland area in 2015. GOBLIN prioritises the sparing of organic soils because of the imperative to rewet these soils in order to mitigate LULUCF emissions. Any spared area that exceeds the area of organic grassland soil is deemed mineral soil by default. The spared organic and mineral soil areas are then assigned various land uses. Drained organic soils are either rewetted or converted to fallow (drainage maintained) depending on scenario input regarding fraction of spared organic soils rewetted. On spared mineral soil areas, the proportion of area afforested is determined by the scenario input values. Area that has not been allotted to afforestation is said to be left in "usable condition". Fig. 4 summarises the apportioning of spared area in GOBLIN.

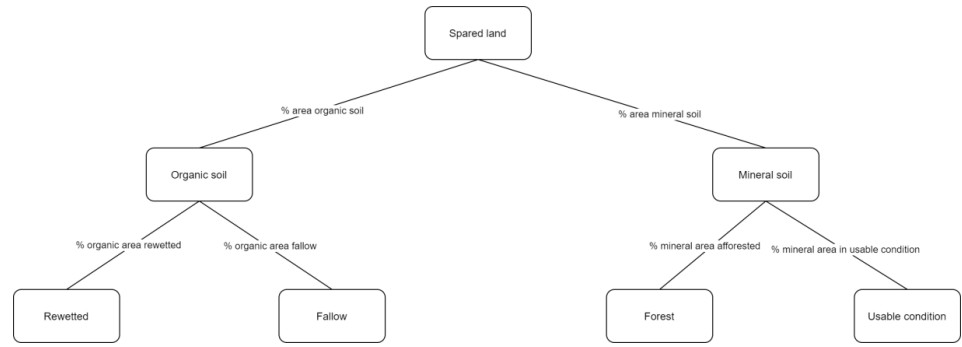

**Figure 4.**         **Allocation of spared land across different primary uses**

### 3.3.2 Forest emissions

Additional land use emissions not accounted for in the forest sequestration module are calculated in the land use module. These emissions relate to drainage and rewetting of organic soils, biomass burning, land use conversion and deforestation. The $CO_2$, $N_2O$ and $CH_4$ emissions from drained organic forest soils and drain ditches are based on the IPCC good practice guidelines (IPCC, 2006) and the 2013 wetlands supplement (Hiraishi et al., 2014). In addition, the NIR (Duffy et al., 2020c) breaks these organic soils into nutrient-rich and nutrient-poor organic soils. The default emission factor of 2.8 kg ha$^{-1}$ yr$^{-1}$N$_2$O-N is applied to nutrient-rich organic soils, however, Duffy et al (2020c) utilise a country specific emission factor of 0.7 kg ha$^{-1}$ yr$^{-1}$ N$_2$O-N on organic soils classed as poor. The $CH_4$ emissions from drained organic soils and drained ditches are also based on default emission factors from the IPCC wetland supplement (Hiraishi et al., 2014) and country-specific parameters were derived from the NIR (Duffy et al., 2020c).


### 3.3.3  Grassland Emissions

Grassland emissions accounted for in the land use module relate to drainage and rewetting of organic soils, biomass burning and land use conversion. A Tier 1 methodology from the IPCC (2006) is utilised to estimate the direct carbon loss from drainage of organic soils. The default emissions factor of 5.3t C $ha^{-1}$ $y^{-1}$ for shallow drained managed grassland soils for cold temperate regions is derived from the 2013 wetlands supplement (Hiraishi et al., 2014). The estimation of emissions from the drained inland organic soils derives from the 2013 wetlands supplement (Hiraishi et al., 2014). The default emission factor of 4.3 kg $N_2O$–N $yr^{-1}$ for nutrient poor, drained grassland from the 2013 wetlands supplement (Hiraishi et al., 2014) is utilised. Tier 1 IPCC (2006) methodology is used to estimate $CO_2$ removals (from the atmosphere) via uptake by soils, $CO_2$ losses from dissolved organic carbon to water, and $CH_4$ emissions. Emissions factors are again derived from the 2013 wetlands supplement (Hiraishi et al., 2014). Finally, emissions of $CH_4$ and $N_2O$ from the burning of biomass are estimated utilising the IPCC (2006) Tier 1 approach.

### 3.3.4  Wetland Emissions

Wetland emissions include $CO_2$ from horticultural peat extraction, drainage and rewetting and burning, $CH_4$ and $N_2O$ from drainage and burning, and $CH_4$ from rewetting. The NIR (Duffy et al., 2020c) includes emissions related the extraction and use of peat products under the category of "horticultural peat". Data related to the quantities of exported peat are reported by United Nations Commodity Trade Statistics Database (UN, 2016). To calculate off-site emissions from peat products, GOBLIN utilises a Tier 1 methodology (IPCC, 2006) to estimate carbon loss by product weight.

Carbon stock changes in biomass are determined by the balance between carbon loss due to the removal of biomass when preparing for peat harvesting, and the gain on areas of restored peat lands (Duffy et al., 2020c). Non-$CO_2$ emissions related to drainage and rewetting are $CH_4$ and $N_2O$. $CH_4$ emissions estimations utilise the methodology provided in the 2013 wetlands supplement (Hiraishi et al., 2014) and require an estimate of the area impacted by drainage and the density of drainage ditches. Annual direct $N_2O$–N emissions from drained organic soils are estimated utilising a Tier 1 approach based on the IPCC (2006) methodology and a default emission factor of 0.3 kg $N_2O$–N $yr^{-1}$.

GOBLIN also calculates emissions from $CH_4$ and $N_2O$ from biomass burningThe value used in the NIR (Duffy et al., 2020c) to represent the mass of fuel available for burning is 336 t $ha^{-1}$ DM. The emissions factor values utilised for $CO_2$, $CH_4$ and $N_2O$ correspond to 362 g $kg^{-1}$, 9 g $kg^{-1}$ and 0.21 g $kg^{-1}$ DM burned, respectively.

### 3.3.5  Cropland Emissions

Cropland emissions are estimated utilising a Tier 1 approach (IPCC, 2006). $CO_2$ emissions include emissions related to land use transitions from grassland or forested land to cropland and from biomass burning. $N_2O$ and $CH_4$ are also related to biomass burning. Emissions of $CO_2$, $CH_4$ and $N_2O$ from the burning of crop biomass are also estimated utilising the IPCC (2006) Tier 1 approach.





### 3.4 Forest management

Irish forest cover accounts for about 11% of total land area (DAFM, 2018). Conifers make up over 71% of the forest estate, the main species being Sitka spruce (*Picea sitchensis (Bong.) Carr.*) (SS) comprising over 50% of total forest land area. In 2017, broadleaf species made up almost 29% of total forest land area (DAFM, 2018; Duffy et al., 2020b, 2020a). However, given that the historic rate of broadleaf inclusion within afforestation was less than 10% for significant periods (DAFM, 2020b), GOBLIN utilises an aggregate value of 20% broadleaf inclusion to represent historic afforestation. Given the complexity in both representing the current forest estate, and simulating future afforestation/reforestation, the forest module is split into two containers: the old forest container (OFC) and the new forest container (NFC). The OFC estimates sequestration from afforestation from 1922 until 2025, and is used to determine the age profile of standing forest. After 2025, the OFC no longer adds area to the model, but continues calculation of growth (carbon sequestration) and harvest (terrestrial carbon removal) in pre-existing forested area until the end of the simulation has been reached (2050 in our example).

From 2025 onwards, sequestration from afforestation is calculated in the NFC utilising annualised afforested areas derived from the target-year spared area calculated in the grassland management model and shares of that area going to forest types (scenario module). The NFC computes sequestration from afforestation from 2025 to the end point of the simulation. The results of the OFC and NFC are added together to calculate total net sequestration in forests. The purpose of this two-step calculation is to save system resources. Net sequestration in the existing forest estate only needs to be calculated once as it remains the same across different scenarios, irrespective of changes in the afforestation rate. As such, we utilise the OFC a single time, adding the static results to the variable output from each scenario generated in the NFC.

Fig. 5 illustrates the flow of data through the forest model. The brown rectangles represent entities, mainly conifer and broadleaf, for old and new forest. The purple circles represent processes, while the green rectangle represents a common data store. The old and new forests are kept in separate containers before being aggregated. To estimate the various elements (sequestration from biomass, organic and mineral soil emissions, dead organic matter, etc.) for the forest estate, a matrix approach is adopted. For each element in the forest model, a value matrix is established based on the age of the forest stand. Stand age is then utilised to establish the total biomass, dead organic matter and emissions from organic soils. Once the final matrix has been established, it is aggregated into a single vector with a single cell per year. At this point, any further annual additions or subtractions that need to be made are factored into the model. For further detail on the calculation of biomass increment, DOM, organic and mineral soil emissions refer to Duffy et al (2020a).




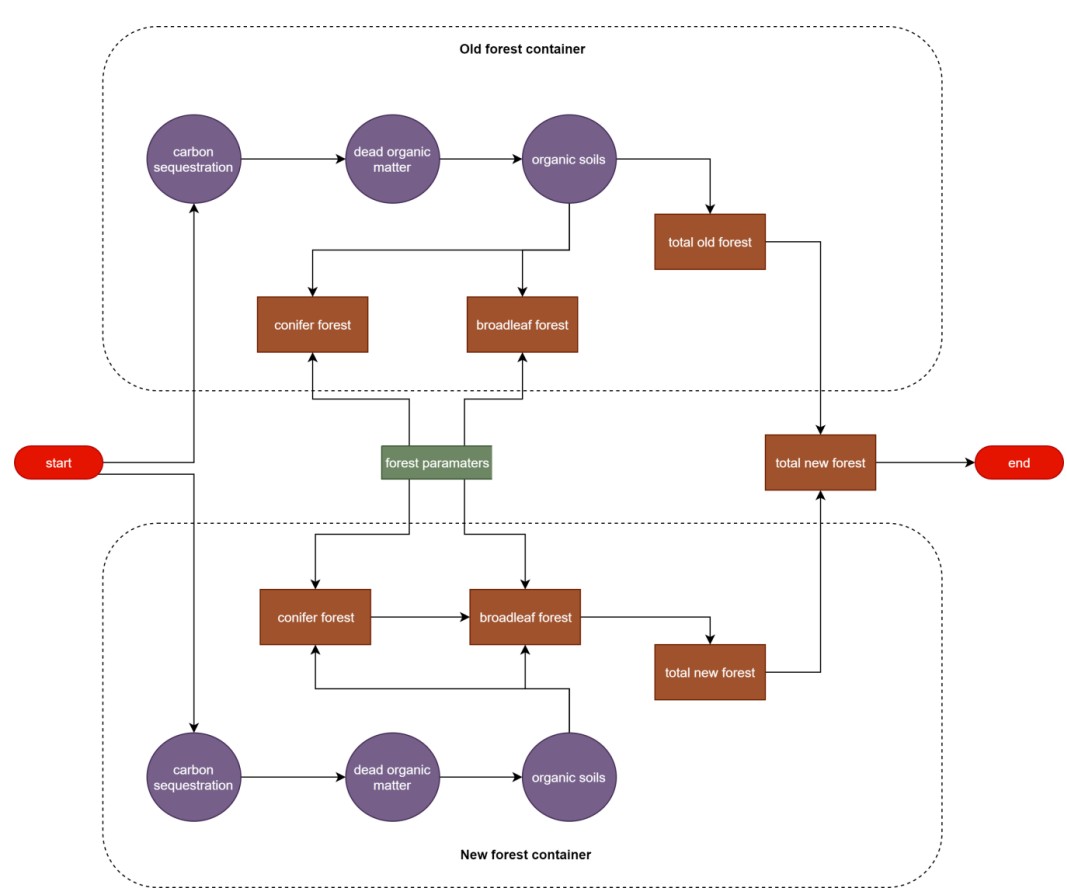

Figure 5. **GOBLIN forest module calculation methodology. Arrows represent data flow. Modules are represented by brown rectangles , processes by purple circles, and open-ended green rectangles represent data stores.**




## 4. Model validation


Validation of emissions computations for livestock production and land use (change) is
achieved by running GOBLIN using the same Central Statistics Office (CSO) activity data
used for NIR activity inputs for a time series between 1990 and 2018. Emissions across all
major sources are then compared between GOBLIN (1990 – 2015) and the NIR (1990 – 2018),
using CRF files dating back to 1990. Fig. 6 and 7 illustrate validation across major emission
sources. Beginning with land use and land use change (Fig. 6), solid lines represent $CO_2$, $CH_4$
and $N_2O$ emissions modelled in GOBLIN, while the dashed lines represents equivalent
emissions reported in the NIR. Absolute emission levels and trends calculated by GOBLIN
very closely match those of the NIR, with the most notable deviation arising for forest
sequestration (representing the complex Tier 3 modelling of fluxes, sensitive to compound
estimates of stand age profiles across hundreds of land parcels). Fig 6. shows validation of
agricultural emission sources. Enteric and manure management $CH_4$ from  GOBLIN and the
NIR are almost identical, while $CO_2$ and $N_2O$ emissions levels and trends are very similar. This
validation specifically indicates that emission factors, land area calculations, forestry
increments and harvest removals, and animal feed intake calculations derived from raw input
data are in line with NIR methodology, providing confidence in scenario extrapolations based
on variations in these input data.

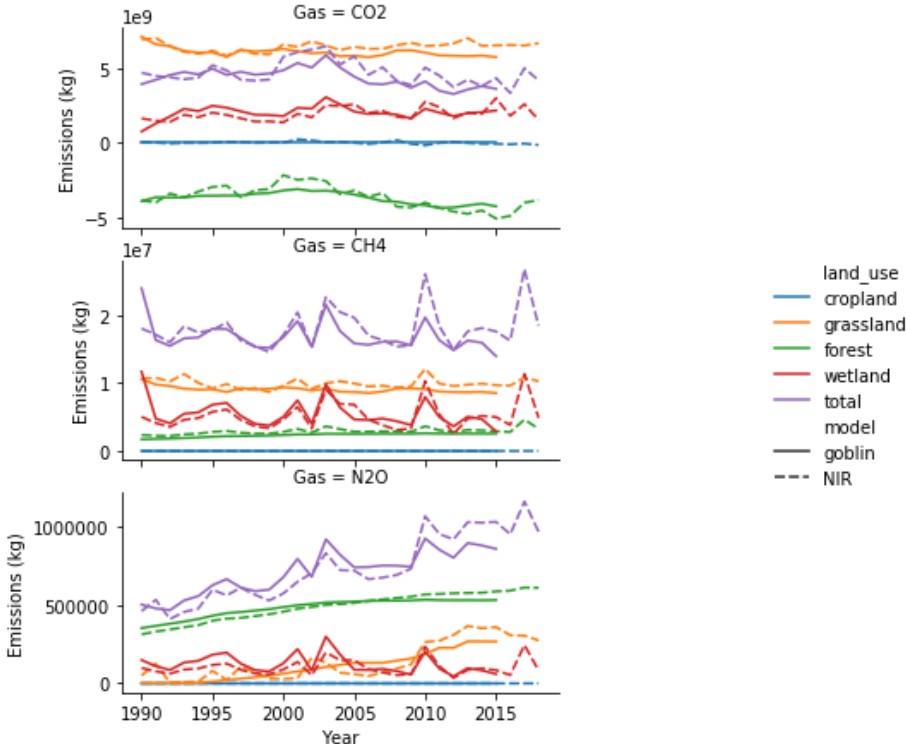


**Figure 6.**       **Comparison of land-use emissions between GOBLIN and the NIR, derived**
528          **from the same activity data for 1990 to 2015**






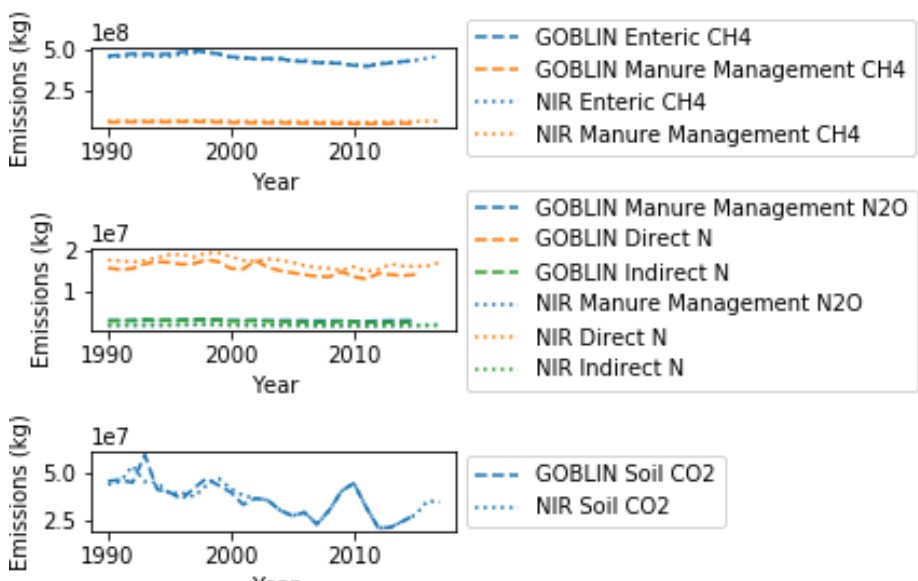


**Figure 7.** **Comparison of agricultural emissions between GOBLIN and the NIR, derived from the same activity data for 1990 to 2015**


## 5. Example of Model Output

To demonstrate and explore the critical functions of GOBLIN, several scenarios were analysed to reflect national level GHG reductions within the AFOLU sector (Table 4). As set out in Ireland's Climate Action Bill (2021), Ireland must achieve a 51% emission reduction by 2030. Given that agriculture makes a significant contribution to the national emissions profile (DAFM, 2020a), the illustrative scenarios produced as part of this model summary reflect potential emissions reduction pathways. In terms of animal numbers, all scenarios reflect reductions in dairy, beef and sheep numbers of 10%, 50% and 50%, respectively, by 2050. In terms of land use, all scenarios, with the exception of scenario 4, assume at least the baseline (recent average) afforestation rate continues to 2050 (the average afforestation rate was 6,664 ha yr$^{-1}$ between 2006 and 2017 (Duffy et al., 2020a)). All annual afforestation rates continue to 2050, with zero afforestation assumed after 2050, and are based on a 70:30 conifer:broadleaf mix.

**Table 4.**   **Summary of indicative scenarios analysed using GOBLIN**

| Num | Description | Details | Afforestation rate (ha per year) |
|---|---|---|---|
| 0 | Animal reduction | • Dairy, Beef and sheep herd numbers reduced by 10%, 50% and 50%, respectively by 2050 <br><br> • Base afforestation rate applied | 6664 |
| 1 | Animal reduction and rewetting | • Dairy, Beef and sheep herd numbers reduced by 10%, 50% and 50% by 2050, respectively. <br><br> • 100% of organic soil under grassland rewetted <br><br> • Base afforestation rate applied <br><br> • Remaining spared land kept in "farmable condition". | 6664 |
| 2 | Animal reduction and afforestation | • Dairy, Beef and sheep herd numbers reduced by 10%, 50% and 50% by 2050, respectively. <br><br> • 100% area mineral and afforested. | 35785 |
| 3 | Animal reduction, afforestation and wetlands | • Dairy, Beef and sheep herd numbers reduced by 10%, 50% and 50% by 2050, respectively. <br><br> • 100% of organic soil under grassland rewetted <br><br> • Remaining area assumed to be mineral and afforested. <br><br> • Remaining organic area taken out of production | 26086 |





| 4 | Animal reduction and increased production | • Dairy, Beef and sheep herd numbers reduced by 10%, 50% and 50% by 2050, respectively.<br><br>• Milk output increased by 14% per cow<br><br>• Beef live weight + 20% | 0 |
|---|---|---|---|
| 5 | Animal reduction, increased, afforestation and wetlands production | • Dairy, Beef and sheep herd numbers reduced by 10%, 50% and 50% by 2050, respectively<br><br>• Milk output increased by 14% per cow<br><br>• Beef live weight + 20%<br><br>• 100% of organic soil under grassland rewetted<br><br>• Remaining area assumed to be mineral and afforested.<br><br>• Remaining organic area taken out of production | 24299 |


Fig. 8 and 9 present the main AFOLU GHG fluxes. Firstly, the agricultural emissions (Fig. 8)
illustrate the results for $CH_4$ emissions from enteric fermentation and manure management,
$N_2O$ results from manure management and other direct and indirect $N_2O$ emission pathways,
and finally, $CO_2$ emissions from fertiliser application to soils Emissions related to livestock are
slightly higher in scenarios that have increased production related to milk and beef output than
scenarios with default production estimates.
Fig. 9 illustrates land use emissions related to $CH_4$, $N_2O$ and $CO_2$. Firstly, we examine $CH_4$
emissions from land use and land use change. The changes relative to the baseline year are as
a result of a decrease in grassland area and changes in forest and wetland areas. Changes in
grassland $CH_4$ results from reduction in animal numbers, rewetting of organic soils and
removal of production from organic soils. Relative to scenario 0, the straight animal reduction
scenario, there is a 19, 20 and 22% increase in $CH_4$ emissions in scenarios 1, 3 and 5,
respectively owing to rewetting of drained organic soils. These increases are largely observed
in the grassland category, with some additional emissions in the forest and wetland categories.
In the wetland and cropland categories, an increase is observed relative to the baseline year.
This is explained by the utilisation of a multi-year average to estimate the burned area, this
average is higher than the baseline year, as such emissions related to burning in the target year
are higher.
Secondly, we examine $N_2O$ emissions related to land use and land use change. Relative to
scenario 0, we can observe a 3-4% increase in emissions for scenarios 1, 3 and 5, respectively.
The increases in emissions from wetland areas are related to the rewetting of previously drained
soils. Again, we can see that cropland emissions seem to increase, however, this is again a
reflection of burned area assumptions. The next noticeable difference is in terms of grassland
$N_2O$ emissions which appear to fall dramatically. Past $N_2O$ emissions in this category are
driven largely by conversion of modest amounts of forested land to grassland. As the model
assumes land is converted from grassland to other uses, and not the other way around, the
emissions in this category drop significantly. Relative to scenario 0, emissions in scenarios



where rewetting takes place increase by 20%. As there are no changes to cropland, emissions remain constant among scenarios, the increase relative to the baseline year is again explained by assumptions regarding the burned area.

Finally, Fig. 9 presents the $CO_2$ emissions from land use change. Emissions related to grassland, relative to scenario 0, drop to less than 0.1% in scenarios (scenarios 1, 3 and 5) where rewetting has taken place. Regarding forestry, Fig. 9 highlights the expected value in 2050, drawing a line linearly from 2015 to 2050. As expected, sequestration potential is greater at higher levels of afforestation. The entire time series is explored in more detail in Fig. 10. Wetland emissions increase, relative to scenario 0, by 4 and 5% in scenarios in which rewetting takes place. Lastly, we have assumed no emissions changes for cropland.

To further elaborate the forestry modelling, Fig. 10 shows the forest sequestration time series for each of the scenarios. As can be seen, scenarios 0, 1 and 4 reflect the average afforestation rate, or the "business-as-usual" land use change, and no afforestation. Scenarios 2, 3 and 5 increase sequestration potential significantly. Scenario 2 assumes that all spared area is on mineral soils and as such this scenario has the highest afforestation rate, and the highest sequestration potential. Scenario 3 assumes that all drained areas are rewetted, and the remaining land area is mineral and afforested. Lastly, scenario 5 assumes the same, however, there is less land area available as a result of increased production output from animals. The time series also inherently factors in the harvesting rates. All scenarios assume that afforestation, if applicable, take place up to 2050, with zero thereafter.



**Figure 8.**    Scenario agricultural CH₄ N₂O & CO₂ emissions from enteric fermentation, manure management, direct and indirect N₂O sources and synthetic fertiliser application to soils



**Figure 9.**    Scenario agricultural CH₄, N₂O  CO₂ emissions/removals cropland, forest, grassland and wetland land uses



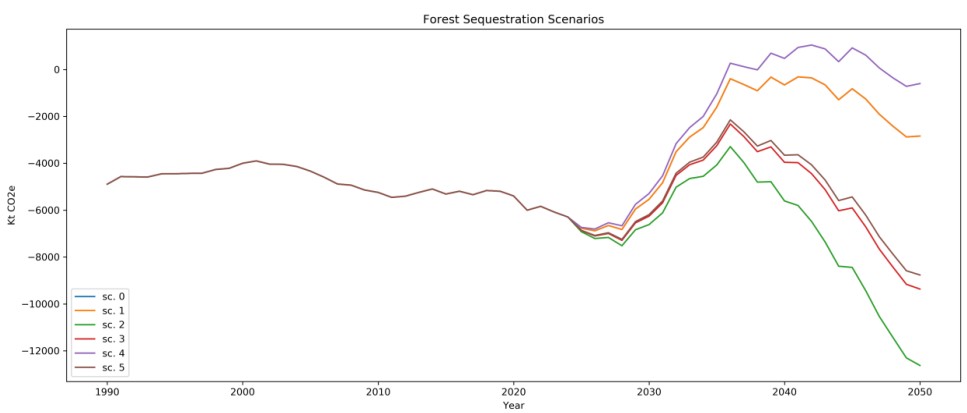

603

**Figure 10.**      **Net marginal (CO₂e emissions accounted for) CO₂e sequestration time series from 1990 to 2050**

604

605



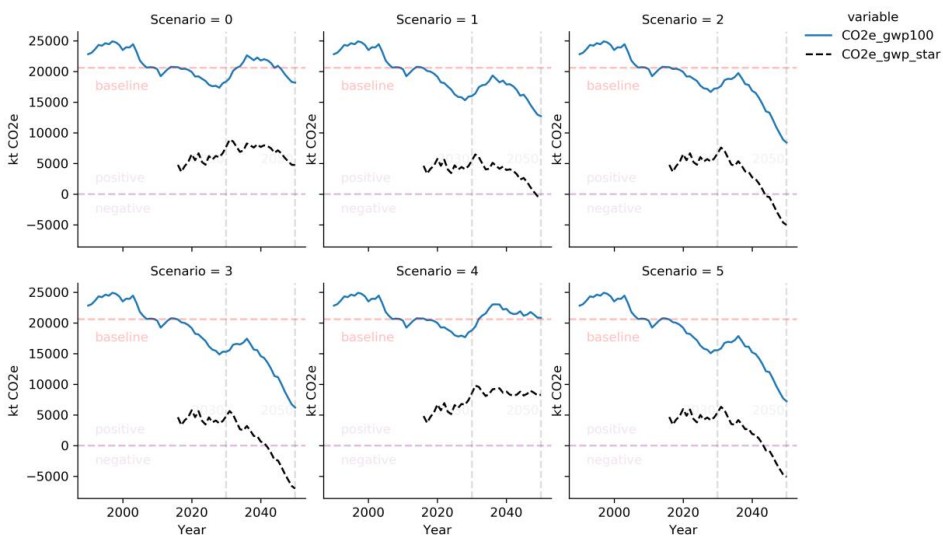

\* blue lines represent GWP100, black line represents the GWP\*.

**Figure 11.     GOBLIN scenario CO₂e aggregation represented in GWP$_{100}$ (blue line) and GWP\* (black line)**

Finally, Fig. 11 represents the aggregated emissions from the AFOLU sector for each scenario using either GWP$_{100}$ or GWP\* to equate warming potential to CO$_{2e}$ emissions. The calculation of GWP\* is based on Lynch et al. (2020). The aggregated emissions are presented net of forest sequestration in order to present a final emissions balance. As can be seen, the reduction in animal numbers drives both emissions reductions. The rewetting of previously drained land provides an easy win in terms of emissions reductions. However, the potential to offset remaining emissions, in terms of carbon sequestration, comes by utilising spared land for afforestation. Both organic soil rewetting and higher rates of afforestation are needed to reduce the GWP$_{100}$ emissions balance, which in the best case (scenario 3) is reduced by circa 73% from the 2015 balance.

## 6.   Forest sequestration time series extension

Fig. 12 presents an extended time series for forest sequestration to 2120. Specifically, Fig. 12 illustrates afforestation to 2050, with 0 afforestation thereafter. A forest conservation approach is considered for all new forest, assuming a 0% harvest rate. This conservation approach does successfully avoid the so called "carbon cliff" in scenarios 2, 3 and 5. However, the marginal gains are reduced over time as trees reach maturity. Ongoing model development will enable longer-term mitigation associated with harvested wood use to be represented.



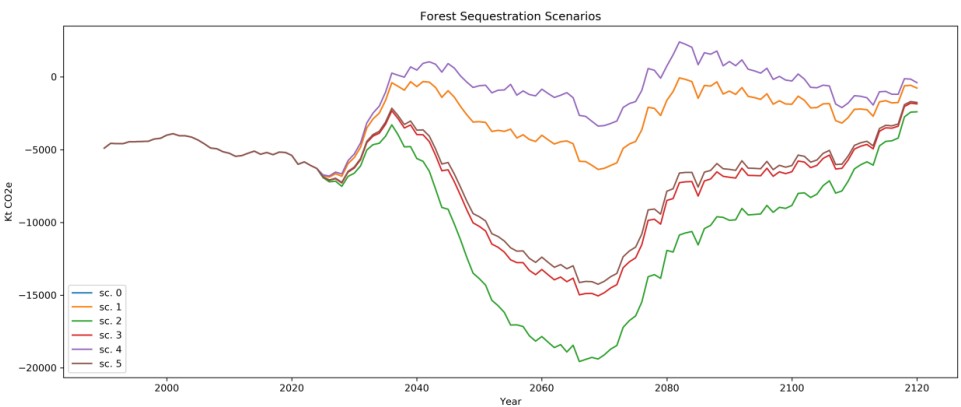

**Figure 12.** Net marginal ($CO_2e$ emissions accounted for) $CO_2e$ sequestration time series from 1990 to 2120 with 0% afforestation post 2050 and 0% harvest rate

## 7. Discussion

### 7.1. National AFOLU models for climate policy

The AFOLU sector is central to global efforts required to stabilise the climate, and will need to shift from being a net source to a net sink of emissions by 2050 in order to constrain anthropogenic global warming to 1.5°C (Masson-Delmotte et al., 2019). Such a shift will require widespread and rapid deployment of appropriate mitigation options to reduce the emissions intensity of agricultural production whilst maintaining food security, alongside food demand management and actions to realise emissions removals via forestry and bioenergy (Huppmann et al., 2018; IPCC, 2019b). The GOBLIN model described here was developed as a tool to quantify long-term (circa 100 year) GHG emission fluxes associated with different AFOLU scenarios representing changes in land use over the next three decades. The intention is to bridge the gap between hindsight representation of national emissions via UN FCCC reporting (Duffy et al., 2020c) and global IAMS models (Huppmann et al., 2018) that are broad in scope but lack (sub)national detail. IAMS global pathways towards climate stabilisation involve many assumptions, and are difficult to downscale to national targets. Whilst a number of countries have set national "net zero" GHG emission targets for 2050 (UK CCC, 2019), there remains considerable uncertainty about the role of distinct national AFOLU sectors, particularly with respect to appropriate targets for $CH_4$ emissions and $CO_2$ offsetting within NDCs (Prudhomme et al., 2021). Ireland provides an excellent case study country to explore possible trade-offs between food production and various definitions of climate neutrality owing to high per capita GHG (including $CH_4$) emissions from AFOLU, both from ruminant food production destined for export and from land management (Duffy et al., 2020c).

GOBLIN has been calibrated against Ireland's NIR (Duffy et al., 2020c) to align outputs with GHG reporting methodologies, but applies a novel land balance approach to determine future combinations of emissions sources and sinks by relating animal feed energy requirements to grass production under different fertilisation and grazing (utilisation efficiency) regimes. Through integration of animal energy demand functions and grass fertiliser response curves, the model is able to vary areas needed to support different combinations of livestock systems at the national level. This functionality enables critical aspects of livestock production



efficiency to be explicitly varied within scenarios, providing deep insight into interactions between livestock production, including sustainable intensification trajectories (Cohn et al., 2014; Havlík et al., 2014) that represent implications for future food production, and biophysically compatible levels of organic soil rewetting and sequestration across forest types. The latter functionality derives from integration of aforementioned livestock system modelling with detailed representation of the complex carbon dynamics of existing and "new" forests. This represents a significant advance in national AFOLU GHG modelling capability, and will build on modelling of livestock emissions displacement with forestry offsets recently calculated in (Duffy et al., 2020a) to provide a solid evidence base for development and implementation of NDCs.

Crucially for a national AFOLU sector so far from complying with any definition of climate neutrality, fully randomised scenario analyses with GOBLIN will generate new evidence on which biophysically coherent combinations of agricultural activities and land uses satisfy particular definitions of climate neutrality. Such a back-casting approach can inform objective comparison of trade-offs, and may also help to elicit more constructive and focussed stakeholder engagement on a complex and sensitive topic. The small number of scenarios modelled in this paper were designed simply to demonstrate the technical potential of the model, but it can be used as a platform to support participatory modelling (Basco-Carrera et al., 2017) or for systematic analysis of alternative land use choices (Loucks and Van Beek, 2017). Combining the biophysical outputs of GOBLIN with socio-economic assessment will be crucial to determine effective climate policy at national level.

### 7.2. Defining "climate neutrality"

When model development began in 2018 it was assumed that achieving "net zero" $GWP_{100}$ balance would be the primary objective for GOBLIN scenario modelling. Such an approach remains valid and in line with UN FCCC reporting, and is applied for other countries' 2050 climate targets (Lóránt and Allen, 2019; UK CCC, 2019). Since then, there has been significant debate about how to combine the short term warming effect of $CH_4$ with the long-term cumulative warming effect of $CO_2$ and $N_2O$ (Cain et al., 2019; Prudhomme et al., 2021). An important but initially unanticipated use of GOBLIN will therefore be to explore the implications of various possible definitions of "climate neutrality", underpinned by different value judgements. It is clear from the small selection of indicative scenarios analysed in this paper that choice of GHG aggregation metric and definition of climate neutrality profoundly alters the mix of agricultural production and land use (change) compatible with climate neutrality in 2050 and beyond. Specifically, a "no further warming" definition, represented by a zero balance for GWP* (Lynch et al., 2020), is achieved (or exceeded) by 2050 among four of the six indicative scenarios explored here, whilst "net zero GHG", represented as a zero balance for $GWP_{100}$ (IPCC, 2013), is not achieved across any of the scenarios by 2050. For example, reducing the dairy herd by 10%, and beef cattle and sheep numbers by 50%, could result in "no further warming" (GWP* balance) climate neutrality in 2050 assuming all organic soils are rewetted and recent rates of afforestation (just under 6,700 ha $yr^{-1}$) are maintained. However, the same scenario brings the AFOLU sector only half way towards net zero GHG emissions ($GWP_{100}$ balance) by 2050. Separate calculation of each major GHG within GOBLIN will enable a wider range of climate neutrality "filters" to be applied beyond these simple GWP balance examples, such as a separate target for $CH_4$ combined with a $GWP_{100}$ balance across $N_2O$ and $CO_2$. Over half of global $CH_4$ emissions come from food production (Saunois et al., 2020); detailed modelling of ruminant food production compatible with various



approaches to determine territorial climate neutrality could contribute significantly to policy formulation on separate $CH_4$ targets, e.g. the EU Methane Strategy.

### 7.3. Model limitations and development priorities

GOBLIN examines rewetting of drained organic soils and forestry as the primary mechanisms of emissions mitigation and offset within Ireland's LULUCF sector, reflecting the "main levers" that can be pulled to achieve climate neutrality. Additional land use-technology interactions that could realise significant GHG mitigation by 2050 include bioenergy crop production, such as willow and miscanthus for electricity, heat or advanced liquid biofuel chains, and manures or grasses for biomethane production (Englund et al., 2020; Van Meerbeek et al., 2019). GOBLIN can be adapted and coupled with existing downstream energy emissions models to explicitly represent AFOLU consequences of such options, as well as to illustrate inter-sectoral mitigation pathways (Fig. 1). In this regard, it is important to note that the forestry element of GOBLIN is relatively sophisticated, representing forest composition in terms of broadleaf and conifer species mixes, differing forest management practises and harvest rates. This provides interesting possibilities to link AFOLU mitigation with future use of harvested wood products, possibly in cascading value chains that store carbon in wood products before end-of-life use for bioenergy carbon with capture & storage (BECCS) that can transform forestry $CO_2$ sequestration into potentially permanent offsets (Forster et al., 2021).One of the first applications of GOBLIN will be to couple AFOLU forestry outputs with downstream LCA modelling of wood value chains in order to generate robust projections of $CO_2$ offsetting out to 2120, providing new insight into the post-2050 longevity of various climate neutrality scenarios. Finally, whilst GOBLIN has been extensively validated against the NIR for current management practises, components such as fertiliser-response curves for grass productivity could be altered by new grass varieties or mixed grass-clover swords, or updated to be more spatially explicit in relation to soil and land categorisations (O'Donovan et al., 2021). There is potential to adapt this (and other) components of GOBLIN to represent specific mitigation options. Acknowledging that there are still important developments related to, *inter alia*, management of harvested wood products and bioenergy production to be included in future iterations of the model, GOBLIN represents a powerful tool for academics and policy makers to better understand what is required to reach climate neutrality within Ireland's AFOLU sector (and indeed other national AFOLU sectors dominated by livestock production). Crucially, GOBLIN decouples scenario generation from preconceptions of what pathways to climate neutrality could look like by enabling randomised scenarios to be generated and filtered in a backcasting approach. Although such modelling on its own cannot provide all the answers, it does establish a range of biophysically plausible targets which stakeholders can select from and chose to navigate towards, considering important factors such as delivery of wider ecosystem services, and socio-economic and cultural feasibility.

## 8. Conclusion

The AFOLU sector is both a source and a sink for GHG emissions. The sector will play a key role in mitigation of emissions via reduced agricultural emissions intensity and increased carbon sequestration and other off-setting/displacement activities. GOBLIN is a high resolution "bottom-up" bio-physical model for Ireland's AFOLU sector. Then novelty of GOBLIN lies in its integration detailed land requirements and GHG emissions associated with different levels of livestock intensification and grassland management on one hand, and sophisticated representation of forestry carbon dynamics on the other, alongside other important land use emission sources and sinks. GOBLIN is aligned with, and validated against,



Ireland's inventory reporting methodology for GHG emissions, including a Tier 2 approach for
livestock emissions and a Tier 3 approach for forestry. By calculating GHG flux trajectories
towards (randomised) future (2050) scenarios of agricultural activities and land use (change),
GOBLIN is able to provide new insight into the biophysical boundaries associated with
different definitions of climate neutrality. This could help ground an increasingly polarised
debate around the role of AFOLU in ambitious national climate policy. Detailed representation
of current and future forestry combinations (species, management and harvesting mixes) also
provides a powerful platform for future downstream modelling of harvested wood product uses
in the bioeconomy. This could be complemented by integration of bioenergy uses for spared
land through further model development and/or coupling with existing bioenergy models, and
will enable the evaluation of long-term (to 2120) GHG fluxes in order to determine more
enduring climate neutrality actions. Following model development and validation, GOBLIN
will be used to provide a unique, impartial and quantitatively rigorous evidence base on actions
and strategies needed to achieve climate neutrality across Ireland's AFOLU sector.

**Code Availability**
The exact version of the model used to produce the results used in this paper is archived on
Zenodo (Duffy et al., 2021) and freely available for download.
**Author Contribution**
Duffy, C conducted design, development, analysis, testing and validation and manuscript
preparation.
Prudhomme, R conducted design, development, analysis and validation.
Duffy, B conducted design and development.
Gibbons J conducted validation, review and editing.
O'Donoghue, C conducted validation, review and editing.
Ryan, M conducted validation, review and editing.
Style, D conducted design, development, analysis, review and editing.
**Competing Interests**
The authors declare that they have no conflict of interest.
**Acknowledgements**
*This research was supported by the Environmental Protection Agency (Ireland) (EPA 2018-CCRP-MS.57).*
*Thank you to the National University of Ireland Galway, University of Limerick and Teagasc for the facilitation*
*of this research.*





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
