# Peer review of "GOBLIN: A land-balance model to identify national agriculture and land use pathways to 1 2 climate neutrality via backcasting 3 Colm Duffy1\*, Remi Prudhomme2\*, Brian Duffy6, James Gibbons3, Cathal O'Donoghue4, Mary Ryan5, David Styl"

_Geoscientific Model Development, 2021_

## Author Response (AR1)

Dear Reviewers, we would like to thank you for you detailed feedback on this piece of work. The tables below attempts to summarise the major concerns that were outlined. We have numbered these, however, they do not completely align with the original numbering as we have broken a few comments into separate responses for clarity within the reference table. We have not included specific responses to the minor comments, these are aggregated in a sperate table.

Each row in the table below includes an issue number, a summarised reviewer issue, a response and action to be taken by the authors.

**Table of Requested Edits: Reviewer 1**

| Reviewer Major Requests | | | |
|---|---|---|---|
| **Number** | **Issue** | **Response** | **Actions** |
| 1 | Principal concerns relate to broader applicability of the model itself or the underlying approach to other jurisdictions where in many cases the land-use considerations will differ quite widely from an Irish context | The model is, with additional contextual parameterisation, transferable to other regions.

Where possible, we have ensured that parameters are not hard-coded and a regionalised database for each of the modules has been used.

We have failed to highlight this sufficiently, and the reviewer makes an important point, there are contexts in which the parameterisation will take significantly less effort. It is worth pointing out that the methodology is Tier 2 (in some cases Tier1) compatible for the most part. The forestry component is Tier 3, and this would take time to adjust to | • Additional section on global transferability |

| | | | |
|---|---|---|---|
| | | differing contexts. However, the framework is built to accommodate this.

The authors propose a section on global transferability potential that will highlight limitations, but also illustrate linkages and insight identified at national level that are transferable to other countries i.e. the model generates new insight into specific trade-offs and complementarities at national level that should stimulate targeted analysis of similar issues in other countries with related land uses). | |
| 2 | The validation is also inherently circular in nature in that validation is performed against the training dataset. | There is a clear need to address the short-comings in our elaboration of the validation and the methodological approach.

The GOBLIN model utilises Emissions Factors that are Irish specific from the National Inventory Report (NIR) but uses independent energy calculation for livestock from the IPCC. It is these livestock numbers that determine the grass area utilised. The forest component is also independently built. | • National Herd validation

• Additional validation elaboration

• |

| | | The intention here was to recreate the emissions calculation functions of the NIR for AFOLU activities, but in a way that allowed for the generation of multiple scenarios for back-casting.

The utilisation of emissions factors specific to the national inventory is necessary, given the objectives of the GOBLIN model.

To assess whether this objective has been met, it is also necessary to utilise real world activity data input from the NIR.

NIR activity-emission relationships through time is an appropriate type of "validation" in terms of the emissions computation and forest model output.

However, there are additional areas of validation that can be expanded upon beyond that of the emissions/removals calculations. The authors suggest additional validation mechanisms related to the extrapolation of the national herd numbers. In addition, further elaboration on the validation to be added in text, | |

| | | as well as a more detailed explanation of purpose and functionality of GOBLIN in the methods and discussion. | |
|---|---|---|---|
| 3 | More circumspect about potential use in other contexts, at least without some pretty substantial modifications being undertaken to the model as it stands. | This links back to issue #1, and can be addressed in the proposed additional section on model global transferability. | • Additional section on global transferability (from issue #1) |
| 4 | It is implied that there is a hard wired condition in the model that neutrality must be reached in 2050? This is implied in several places and would constitute a major limitation for its universal application where different jurisdictions may wish to set earlier or later dates for a condition of neutrality in LULUCF to be reached consistent with their NDCs | This is a misunderstanding, which we must address in the text. Identifying climate neutrality pathways was the motivating factor for development of GOBLIN, and is an important use.

However, the "target year" is adjustable by the user. The only reason that 2050 was selected in this illustrative piece is due to the climate objectives within the Irish context. This is not a limitation, and flexibility is build-in.

Further, randomised simulation model means it will generate many non-neutral scenarios that can then be objectively screen according to different criteria (e.g. climate neutrality definitions) | • Elaboration of adjustable "target year" in the methodology |

| | | | |
|---|---|---|---|
| 5 | For international applications it is unclear to what extent a number of the parameters in the model are specifically hardwired to the Irish context. Be explicit in this regard. | This links to issue #1 and can be addressed as part of the proposed new section. | • Table to be included as part of the new section on global transferability |
| 6 | Some of the assumptions seem a little ad hoc. For example, the fertiliser leaching is assumed to be 10% in line 378. Is it correct to infer this is a fixed assumption in the model? If so presumably the model is underdispersive? It would be important to note such limitations comprehensively. | This is a weakness linked with the NIR, and explicit acknowledgment is necessary. This can be addressed in future versions of the model utilising an N-balance approach to leaching. | • explicit acknowledgment in text |
| 7 | The land-use allocation module is highly optimised to an Irish context. For global applicability it would be required to apply numerous additional module features presumably? | Again, this can be acknowledged in the new section dealing with issue #1. Limitations and scale of reparameterization necessary can be highlighted there. The level of reparameterization necessary will depend on the departure from the Irish context. However, the most important aspect will be highlighting any fixed assumptions. | • Table to be included as part of the new section on global transferability |
| 8 | In all figures careful attention is required regarding the font size – in many figures the font is illegible in the printed copy owing to small font sizes | Noted | • Adjust figure font size |
| 9 | In the model validation piece, the NIR numbers are taken as 'truth' but in reality these are highly uncertain. What danger is there of overtuning having occurred whereby if the NIR numbers are wrong then so is the GOBLIN model output? | The purpose of GOBLIN was to contribute to the policy context within Ireland, so there was not getting away from the fact that NIR data had to be utilised to some degree | • Acknowledge QA/QC procedures in text |

| | | | | |
|---|---|---|---|---|
| | | regarding the validation approach. However, this makes the model a valuable policy tool.

In terms of accuracy, NIR is subject to external and internal review. There is a detailed QA/QC procedure in place. External reviews of the agriculture sector and the entire ETS have been conducted involving both the department of agriculture and, in a separate bilateral review, UK agriculture experts. In addition, the transparency, robustness and accessibility of the inventory data was assessed by Aether (environmental data specialist). | | |
| 10 | | would expect more on validation and a more critical assessment of the suitability of NIR numbers for the task. | Agreed that more detail on limitations and a greater degree of consideration regarding validation is necessary. Additional validation have been elaborated in #2.

However, given the purpose of the model, and the fact that the NIR have quality control and assurance procedures, and are audited, the authors would assert that country specific factors are appropriate. | • Additional detail on validation approach and suitability in text |

| 11 | scenarios used in Figures 8-12 and associated text short names rather than using numerical identifiers for ease of reader comprehension. | Noted | • Numbers replaced with short-names |
|---|---|---|---|
| 12 | It is very clear from Figures 8-10 but particularly 8 and 9 that the GOBLIN model fails to capture real-world interannual variability yet this goes unremarked. This would raise concerns in readers minds as to the veracity of the model. | The author is correct that the projection to 2050 does not include the interannual variation generated by exogenous factors. This will be explicitly noted. The suggestion of a stochastic noise generator can also be considered for future iterations. | • Explicit acknowledgement re interannual variation |

**Table of Requested Edits: Reviewer 2**

| Reviewer Major Requests | | | |
|---|---|---|---|
| **Number** | **Issue** | **Response** | **Actions** |
| 1 | The paper lacks a discussion of the model framework chosen, and it does not put this in the context of existing Integrated Assessment Models (IAMs) or other model types | The suggestions made by the reviewer are noted and an additional section developing the modelling context will be added. | • Additional section on modelling context |
| 2 | To date, the main GHG mitigation measured proposed by the Irish agri sector is improving production efficiencies (genetics, protected urea, multi-species swords, feed additives etc) e.g., represented in the Teagasc MACC. It is not clear to what extent these are taken into account in the input parameters. | The input parameters that are varied in the current version of GOBLIN are explicitly mentioned, but perhaps this has not been made clear enough in the text. In terms of productivity increases, some efficiency gains are implicitly accounted for through the | • Detaill impact of future research on EFs

 • Elaborate on the inclusion of levels of technical abatement |

| | | inclusion of beef and dairy productivity increases. |
|---|---|---|---|
| | | However, the addition of explicit MACC technologies has not been included in this as it would seem more prudent to include differing level of technical abatement, rather than specific technologies. | |
| | | However, there is additional work on-going that will produce additional emissions factors (such as grass-clover sward research), The potential of this research can be elaborated on in text. In addition, greater detail on the approach taken regarding the inclusion of levels of technical abatement can also be elaborated on. | |
| 3 | the way the results and scenarios are presented read far more as a simulation tool rather than backcasting. | This point is noted and additional clarifications will be added. The GOBLIN calculation engine is a tool that allows for simulation of outcomes based on parameter inputs. However, the randomisation of those parameter inputs enable back-casting by filtering target-oriented outcomes.

Multiple pathways to comply with biophysical climate | • Additional clarification in text. |

| | | neutrality provide input for next set of socio-economic screening by stakeholders (i.e. a single answer form the model is not desirable). The model also generates nutrient loss results that can be compared. Thus, randomised simulation modelling provides a richer, non-biased dataset of potentially climate neutral scenarios which can then be further analysed by stakeholders using different criteria and potentially parallel (e.g. economic) analyses. | |
|---|---|---|---|
| 4 | Given that land use is not only fundamentally important for carbon sinks and food production; it is also essential to host and enhance biodiversity. Ireland has declared a biodiversity crisis and it is not sufficient to deal with climate change without also dealing with the very poor quality of biodiversity. Some forms of grazing (more extensive systems) are compatible with greater biodiversity, like some forest models. I think it is important for a land use model to work towards explicit incorporation of biodiversity, otherwise there is a risk that climate and food production will come at a cost to nature. Similarly, an explicit output of the model could be nitrate runoff, to highlight water pollution. | This point is well made, however, the objective for this first version of GOBLIN was the generation of various land use pathways that have the capacity to reach "net zero". Once this data set has been generated, they can be assessed in terms of additional impact pathways. Though this additional analysis was beyond the scope of this first iteration of GOBLIN, the quantification of additional impact pathways (including biodiversity) is already being explored for future iterations. A clearer explanation of this current limitation, and future research and development area will be added in the text. | • Additional information in text |

| | | It should also be noted that, though not explicitly focused on within the model, enhanced biodiversity outcomes could be inferred from scenarios that have higher proportion of native broadleaf species on mineral soils and scenarios with a greater proportion of drained organic soil rewetting. However, the actual biodiversity benefits are not currently quantified within this version of the model.

In relation to water quality, GOBLIN treats N inputs using same functions as NIR, assuming c.10% lost to waters. This can be improved in future e.g. using a per hectare mass-balance approach. This functionality will be used when interpreting future outputs. | |

**Table of Minor Requested Edits:**

| Reviewer aggregated minor requests | | |
|---|---|---|
| **Number** | **Issue** | **Response** |
| 1 | Line 90 mentions the need (to -> the) | edited |
| 2 | Line 103 also a net source | edited |
| 3 | Line 120 capitalise one of the n's in in intensification for the acronym to make sense – probably the final one. | edited |
| 4 | The sentence starting line 146 is an odd way to start a paragraph and also leaves open whether neutrality in 2050 is a hard-wired assumption in the model or something that can be varied. The final parentheses also make little logical sense. I would suggest completely redrafting this passage to provide a clearer entry to this paragraph. | redrafted |
| 5 | Line 203 makes no sense – upland and lowland were numbers – do you mean ewe numbers? | edited |
| 6 | The font size in figure 2 should be made larger for legibility | Graph adjusted |
| 7 | The references in the forestry module entry in table 1 seem a little odd. Why in each case is the 'author' stated twice? | edited |
| 8 | Line 259 – will be? Surely this should instead be was? | edited |
| 9 | Table 2 caption should be clear that these are the values appropriate for an Irish application, surely? | edited |
| 10 | Figure 3 again would benefit from larger font size for legibility | Graph adjusted |
| 11 | Line 355 is based upon a methane … | edited |
| 12 | Figure 4 again the text font size needs to be larger for legibility. | Graph adjusted |
| 13 | Line 458 issue of two sentences merged | edited |
| 14 | In Table 4 the last entry 'name' makes no logical sense | edited |
| 15 | Line 722 missing space between sentences | edited |
| 16 | Line 746 The novelty (the not then | edited |

| 17 | L23 AFOLU - Agri, Forestry, Other Land Use | edited |
|---|---|---|
| 18 | L31 specify the year and check. Agri emissions only in 2018 accounted for more than 34% so I would expect that including LULUCF would be more. | edited |
| 19 | L75 - a literal reading of Article 4 suggests emissions are balanced - this can be interpreted as climate neutral only if it's intended that a "removal" of methane includes it oxidisation in the atmosphere. But then it must be clear that according to mitigation modelling "climate neutrality" is not sufficient to meet the Article 2 temperature goals, and ethical issues about how countries seek to achieve these goals remain. | The decision was made not to change the text because this is a complex issue and the main purpose of GOBLIN is to generate emissions time series that can be filtered according to post-hoc definitions of climate neutrality based on these different concepts. We are not prescriptive here to avoid complication, and leave those deliberations to future papers. |
| 20 | L90 "the need to" | edited |
| 21 | L90 - food security: I question whether this provision in PA is meant to safeguard BAU food production of emissions-intensive foods in high-income countries. This is used by corporate lobby groups to excuse the need for mitigation so if it is used in this article. I suggest more discussion and nuance. | Though the reviewer is correct, in that this provision is misused, the context here does not endorse any BAU approach, and is utilised only to add to considerations. |
| 22 | L101 - specify the year - there is more recent data from 2020 EPA accounts - and I suggest citing the original source rather than the author's work. | Year added, this, however, is the original source, the fact that the authors share the same name is coincidental. |
| 23 | L102 - what is the share of AFOLU in overall emissions and how does that compare to other countries? | Additional text added |
| 24 | In the discussion of metrics it would also be beneficial to summarise LULUCF accounting - "gross net" "net net" etc. Current EU land use account for example considers Irish LULUCF to be a sink/credit, which brings some confusion. | Additional text added to the "Model classification, scope & description" section. Line 158:160 |
| 25 | L116 - what is meant by "stakeholder visions"? | There are multiple stakeholders from disparate contexts with vastly differing priorities, each having a an idealised vision of what the future of the Irish AFOLU sector should be. |
| 26 | Fig 1: please increase the text size. No methane? Or is CO2 meant to be CO2e or GHG? | Corrected |

| | | |
|---|---|---|
| 27 | L168 - the interpretation of PA as climate neutrality as cumulative warming over the second half of the 21st century is new to me (but I am not expert in this) | The PA states we should "achieve a balance between anthropogenic emissions by sources and removals by sinks of greenhouse gases in the second half of this century". It doesn't explicitly state this objective is cumulative, but the implication is that it will be an average over time ("time integrated") that could also be represented as a cumulative balance. |
| 28 | Fig 2 - increase text size please for legibility. | Graph adjusted |
| 29 | Table 2 - Dairy cow numbers appear to be too low: https://www.farmersjournal.ie/100-000-lift-in-cow-numbers-forecast-by-2025-655485 | The numbers in Table 2 relate to milking cows only. We acknowledge that numbers are forecast to increase beyond this range, but trajectories required for climate neutrality are almost certainly downwards.

Further, the upper and lower bounds are adjustable, this is not hardwired into the model. |
| 30 | The fact that cropland values are static should be explored in the discussion and possibly in future iterations of the model. Most crops are used for animal feed. I understand that future iterations will consider imported feed also so an future key parameter could be the share of animal feed from domestic vs imported. | Additional text added. Line 793:796 |
| 31 | "The proportion of grass production consumed by livestock via grazing and feeding on conserved grasses (silage and hay)." this is not clear to me. What is done with the remaining grassland? | This refers to the amount of grass produced on a per hectare basis. A large part of grass production, and potential grass production, is not consumed because grazing is not tightly managed. So, grass simply senesces and decomposes at the end of the season without being eaten. |
| 32 | L204 - production intensity is based on national averages. This could be addressed in subsequent model iterations. The emissions intensity of farms varies widely; reducing production on more emissions-intensive systems would be low-hanging fruit. | This is an important point that will be addressed with higher resolution models in future projects. |
| 33 | L280/Table 3 - what do these coefficients relate to? | These coefficients are utilised to compute the population cohort size from the mature dairy and suckler numbers. |
| 34 | Fig 3 - increase font size. Including units in different parts of these graphs would be helpful for understanding, and distinguishing what is an input variable. This is also the case for equations such as eq (2). | Graph adjusted |

| | | |
|---|---|---|
| | | |
| 35 | L370: this line suggests that LU emissions from soil (e.g., drained organic soil) are not included here but the following paragraph suggests it is. | Organic soils emissions are calculated in the LU module. Edited are of confusion. |
| 36 | Fig 4. Increase font size please | Graph adjusted |
| 37 | Fig 8&9 - these can be developed to greatly aid understanding. it is not possible to see which scenarios are "hidden", for example, the font size is too small and colours hard to distinguish. Axes units (0.0000005) not easy to understand and charts in fig 9 are very compressed. | Graphs have been adjusted somewhat to aid readability; however, it is difficult to display these with greater clarity with out breaking up the set into individual graphs. We have elected not to do this given the volume of graphs already presented. |
| 38 | L604 "Net marginal (CO2e emissions accounted for) CO2e sequestration time series from 1990 to 2050" - explain (related to comment on LULUCF emissions accounting). How is this related to Fig9? | This graph is output from related to forest emissions/removals. They are incorporated into previous graph, however, to appreciate the full complexity of forest sequestration over time, it is important to show the full time series. |
| 39 | Fig 11 - Paul Price's work has suggested that cumulative GWP* and GWP100 gives a more accurate representation of ongoing warming impacts. A discussion on this would be beneficial. | Additional text line 773:4 |
| 40 | Important to note that none of these scenarios meets climate neutrality in the conventional GWP100 sense. | Additional text line 759:60 |
| 41 | Results section: it would be very valuable to include outputs of food production (litres of milk etc) and total land use (total share of land under grass, conifer, etc) given that the model is presented as a tool for assessing trade-offs between food production and mitigation, and nitrate runoff to reflect water quality. | This is noted and is incorporated as an important output from the forth-coming scenarios publication. As that paper will deal with analysis of the scenarios generated, this paper illustrates the methodology utilised. |
| 42 | L642: Simply IAMs not IAMS models | edited |
| 43 | L707: Solar and wind renewable electricity also require land use.

This section is quite dense: a list of development priorities would be beneficial. | Some modification of list to indicate that it is not exhaustive. |
| 44 | References - many do not have years | All references contain years, there may be some slight confusion here regarding the journal citation style, which appends |

| | | the year to the end of the reference, instead of after the authors. |
|---|---|---|

---

## Author Response (AR2)

Dear Dr. Huppmann,

Minor corrections and spelling error checks have been made, references double checked. In addition, the line numbers have been removed. Requested figure edits have also been made. The requested folders and checkpoints have also been removed, and the DOI has been updated (updated in references also).

Many thanks for your assistance and support.

Best

Colm Duffy